# Lead exposure is associated with functional and microstructural changes in the healthy human brain

Hikaru Takeuchi [1✉], Yasuyuki Taki[1,2,3], Rui Nouchi[4,5,6], Ryoichi Yokoyama[7], Yuka Kotozaki[8],
Seishu Nakagawa[9,10], Atsushi Sekiguchi[2,11], Kunio Iizuka[12], Sugiko Hanawa[9], Tsuyoshi Araki[13],
Carlos Makoto Miyauchi[6], Kohei Sakaki[6], Takayuki Nozawa [14], Shigeyuki Ikeda[15], Susum Yokota[16],
Magistro Daniele [17], Yuko Sassa[1] & Ryuta Kawashima[1,6,9]

Lead is a toxin known to harm many organs in the body, particularly the central nervous system, across an individual's lifespan. To date, no study has yet investigated the associations between body lead level and the microstructural properties of gray matter areas, and brain activity during attention-demanding tasks. Here, utilizing data of diffusion tensor imaging, functional magnetic resonance imaging and cognitive measures among 920 typically developing young adults, we show greater hair lead levels are weakly but significantly associated with (a) increased working memory-related activity in the right premotor and pre-supplemental motor areas, (b) lower fractional anisotropy (FA) in white matter areas near the internal capsule, (c) lower mean diffusivity (MD) in the dopaminergic system in the left hemisphere and other widespread contingent areas, and (d) greater MD in the white matter area adjacent to the right fusiform gyrus. Higher lead levels were also weakly but significantly associated with lower performance in tests of high-order cognitive functions, such as the psychometric intelligence test, greater impulsivity measures, and higher novelty seeking and extraversion. These findings reflect the weak effect of daily lead level on the excitability and microstructural properties of the brain, particularly in the dopaminergic system.

[1] Division of Developmental Cognitive Neuroscience, Institute of Development, Aging and Cancer, Tohoku University, Sendai, Japan. [2] Division of Medical Neuroimaging Analysis, Department of Community Medical Supports, Tohoku Medical Megabank Organization, Tohoku University, Sendai, Japan. [3] Department of Radiology and Nuclear Medicine, Institute of Development, Aging and Cancer, Tohoku University, Sendai, Japan. [4] Creative Interdisciplinary Research Division, Frontier Research Institute for Interdisciplinary Science, Tohoku University, Sendai, Japan. [5] Human and Social Response Research Division, International Research Institute of Disaster Science, Tohoku University, Sendai, Japan. [6] Department of Advanced Brain Science, Institute of Development, Aging and Cancer, Tohoku University, Sendai, Japan. [7] School of Medicine, Kobe University, Kobe, Japan. [8] Division of Clinical research, Medical-Industry Translational Research Center, Fukushima Medical University School of Medicine, Fukushima, Japan. [9] Department of Human Brain Science, Institute of Development, Aging and Cancer, Tohoku University, Sendai, Japan. [10] Division of Psychiatry, Tohoku Medical and Pharmaceutical University, Sendai, Japan. [11] Department of Behavioral Medicine, National Institute of Mental Health, National Center of Neurology and Psychiatry, Tokyo, Japan. [12] Department of Psychiatry, Tohoku University Graduate School of Medicine, Sendai, Japan. [13] ADVANTAGE Risk Management Co., Ltd., Tokyo, Japan. [14] Research Institute for the Earth Inclusive Sensing, Tokyo Institute of Technology, Tokyo, Japan. [15] Department of Ubiquitous Sensing, Institute of Development, Aging and Cancer, Tohoku University, Sendai, Japan. [16] Division for Experimental Natural Science, Faculty of Arts and Science, Kyushu University, Fukuoka, Japan. [17] Department of Sport Science, School of Science and Technology, Nottingham Trent University, Nottingham, UK. ✉email: takehi@idac.tohoku.ac.jp

Lead is a toxin known to harm many organs in the body, particularly the central nervous system, across an individual's lifespan[1]. Numerous studies have been conducted to determine the effect of lead toxicity on cognitive measures, and several meta-analyses have established that exposure to even low levels of lead may be associated with lower intelligence[2], conduct problems[3], and symptoms of attention-deficit hyper active disorder such as inattention and impulsivity[4]. Many studies have also determined that low-level lead exposure may be associated with lower attention, lower executive function, lower language function, and greater depressive moods for review, see ref. [5].

Basic neuroscientific studies have revealed the diverse mechanisms through which lead disrupts neural systems. For example, lead substitutes for calcium and inappropriately triggers processes that rely on calmodulin[6], which, in turn, trigger a wide range of mechanisms that disrupt neural systems, such as synapse formation, axon dendritic extension, and plasticity. Lead also disrupts neurotransmitter release and neurotransmitter-related systems, particularly those of dopamine[7]. Specifically, lead enhances spontaneous neurotransmitter release and inhibits stimulated neurotransmitter release[8]. Lead has been shown to disrupt the gamma-aminobutyric acid (GABA) pathway and decrease GABA release[9]. Lead is taken up by mitochondria, causing swelling and functional disruptions; these effects lead to cell apoptosis and transform ordinarily benign synaptic transmission mediated by glutamate into excitotoxicity, which damages neurons[10]. Lead accumulation also results in reduced glutamine synthetase activity, which, in turn, causes glutamate accumulation and leads to excitotoxicity[10]. Lead alters white matter via the expression of genes essential to myelin formation[11], delays myelin accumulation[12], and alters the structure of myelin sheaths[13].

Previous neuroimaging studies investigated the effects of lead exposure on neural mechanisms. Stewart et al.[14] revealed the association of higher tibia lead levels with more white-matter lesions and smaller total brain volumes in older former organo-lead workers ($N = 532$). Data from the Cincinnati Lead Study revealed that young adults with higher blood lead levels in early childhood show lower regional gray-matter volume in various areas ($N = 157$)[15], reduced fractional anisotropy (FA) throughout the white matter, increased or decreased mean diffusivity (MD) of the white matter depending on the region ($N = 91$)[16], decreased N-acetyl aspartate in the gray matter (interpreted as neuronal dysfunctions), and decreased choline in the white matter (interpreted as alterations in myelin architecture; $N = 159$)[17]. Data from the same cohort further revealed that young adults with higher lead levels in early childhood show diminished brain activity in the left frontal cortex and increased brain activity in the right frontal cortex during a language task ($N = 42$). Marshall et al.[18] revealed that children of lower-income families living in areas with high risk of lead exposure show lower cognitive test scores and cortical volumes in a cross-sectional study of 9712 children.

Despite these and numerous other psychological studies and multiple meta-analyses demonstrating the toxic effects of lead on behavioral measures, no study has yet investigated the associations between body lead level and the microstructural properties of gray-matter areas. MD of diffusion tensor imaging (DTI) has revealed that the free diffusion of water molecules and various tissue components, including synapses, capillaries, macromolecular proteins, the shape and number of neurons or glias, myelin properties, membranes, and axons[19,20], could lower MD. Our recent review indicated that MD in the dopaminergic system (MDDS) may be characteristically associated with altered dopaminergic system properties, including neural plasticity and neurological states[21]. MDDS is highly negatively correlated with dopamine synthesis capacity, as measured by positron emission

tomography (partial correlation coefficient ≈ 0.7)[22]. Although the abovementioned study investigated the association between lead level and brain function during a language task, it only used a small sample size ($N = 42$) and a cluster size test called Monte Carlo simulation, which is known to show inflated false positive results[23]. To date, no study has yet revealed the accurate association between body lead level and brain activity during attention-demanding tasks. Considering this knowledge gap, the purpose of the present study is to investigate the associations between body lead level and MDDS and brain activity during attention-demanding tasks using a large sample size and robust statistics.

We assessed brain activity during the n-back working memory task by using functional magnetic resonance imaging (fMRI) and MD of DTI, together with FA measures and hair lead levels, in a large sample of young adults. We used the permutation test, which has been shown to control for false positives robustly, for multiple comparison corrections. We also utilized a wide range of cognitive measures to confirm whether the previously reported associations of lead with cognitive measures could also be observed in our sample and reveal the nature of cognitive correlates of hair lead levels.

We hypothesized that hair lead levels are associated with alterations in the MD of gray-matter areas, especially in the dopaminergic system (besides previously observed reductions in FA and increases in MD in white-matter areas), hyperactivation, and lower deactivation during attention-demanding cognitive tasks. Our hypothesis is based on the observed effects of lead on dopamine release, as well as brain excitatory and inhibitory systems, such as those mechanisms related to glutamate and GABA.

Our results show that greater hair lead levels are weakly but significantly associated with (a) increased working memory-related activity in the areas that are recruited during working memory, (b) lower FA in white-matter areas, (c) lower MD including MDDS and greater MD in some parts of the brain, (d) lower performance in tests of high-order cognitive functions, such as the psychometric intelligence test, greater impulsivity measures, and higher novelty seeking and extraversion, which are relevant to dopaminergic functions. Our findings reflect the weak effects of daily lead level on the excitability and microstructural properties of brains, particularly in the dopaminergic system.

## Results

**Basic data**. The mean, standard deviation, and range of raw hair lead levels were 396.65 559.70, and 16.45–6162 ppm, respectively. The distribution of hair lead levels between men and women is presented in Supplementary Fig. 1.

No significant difference was found in logarithmic hair lead levels between men and women ($p > 0.1$).

**Associations of hair lead levels with cognitive measures**. After correcting for confounding variables and multiple comparisons, greater hair lead levels were significantly associated with higher extraversion, novelty seeking, and impulsiveness scores and lower cognitive reflectivity–impulsivity scores (indicating greater cognitive impulsivity), complex arithmetic task performance, total scores of TBIT (intelligence task consisting of simple and complex speeded tasks), and reading comprehension task scores (Table 1 and Fig. 1a–f). Part correlation coefficients for all of the significant associations were <|0.14| and the effect sizes were weak (Table 1).

*Associations of hair lead with FA and MD*. FA and MD analyses were performed with the data from 919 subjects after excluding data, which had artifacts.

**Table 1 Statistical results (beta value, *t* value, uncorrected *p* value, *p* value with FDR[a] correction) of multiple regression analyses using psychological variables and lead levels after correcting for confounding variables.**

| Dependent variable | | | | Lead level | |
|---|---|---|---|---|---|
| | N* | Part correlation coefficient (95% CI) | t | p (uncorrected) | p (FDR)** |
| RAPM[b] | 920 | −0.047 (−0.111 to 0.018) | −1.425 | 0.5275 | 0.4062 |
| Total intelligence score of TBIT[c] | 843 | −0.073 (−0.138 to −0.007) | −2.182 | 0.0338 | 0.049995 |
| Simple arithmetic | 661 | −0.040 (−0.115 to 0.035) | −1.047 | 0.2958 | 0.2440 |
| Complex arithmetic | 661 | −0.089 (−0.164 to −0.013) | −2.292 | 0.006 | 0.0139 |
| Reverse Stroop interference | 918 | −0.053 (−0.117 to 0.012) | −1.596 | 0.0567 | 0.0728 |
| Stroop interference | 920 | 0.014 (−0.051 to 0.079) | 0.432 | 0.6333 | 0.4572 |
| Reading comprehension | 837 | −0.068 (−0.135 to 0.001) | −1.943 | 0.0303 | 0.049995 |
| S-A creativity test | 920 | −0.048 (−0.111 to 0.017) | −1.451 | 0.2 | 0.21 |
| Digit-span | 915 | −0.026 (−0.089 to 0.039) | −0.776 | 0.2274 | 0.2189 |
| POMS vigor | 909 | 0.048 (−0.016 to 0.112) | 1.475 | 0.0402 | 0.0580 |
| Novelty seeking | 919 | 0.132 (0.067 to 0.194) | 4.051 | <1/5000 | 0.00077 |
| Impulsiveness | 919 | 0.124 (0.059 to 0.186) | 3.785 | <1/5000 | 0.00077 |
| Extraversion | 920 | 0.104 (0.040 to 0.166) | 3.204 | <1/5000 | 0.00077 |
| Cognitive reflectivity–impulsivity | 918 | −0.074 (−0.137 to −0.009) | −2.238 | 0.006 | 0.0139 |
| External Preoccupation (score) | 920 | −0.052 (−0.116 to 0.013) | −1.576 | 0.2551 | 0.2266 |
| Beck Depression Inventory | 917 | −0.055 (−0.119 to 0.011) | −1.641 | 0.1441 | 0.1664 |

*The reason why the substantial portion of the subjects in the study have missing data in some of the analyses of cognitive measures (totalintelligence score of TBIT, arithmetic tasks, reading comprehension) is because, in this long project, the measures that were gathered from subjectschanged due to the limitation in the test time and many research purposes that are not related to hair analyses. When there are missing data upto several subjects, then the reason is due to the misunderstanding of the rules despite administration of tests or failure to provide accurate answers to the questionnaires.

**Some uncorrected p values are greater than the p values corrected for FDR. The latter are indeed correct. In some FDR methods, including the one used in this study, the phenomenon of corrected statistical values exceeding the original p values) can occur when some p values among the group of analyzed p values are very low. This phenomenon is described in ref. [94]).

[a]False discovery rate.

[b]Raven's advanced progressive matrices (a general intelligence task).

[c]Tanaka B-type intelligence test.

Whole-brain multiple regression analysis showed that hair lead levels are significantly and negatively associated with FA in white-matter areas of the anterior and posterior limb of the right internal capsule, right superior corona radiata, and superior fronto-occipital fasciculus (Fig. 2a, b and Table 2).

Whole-brain multiple regression analysis also showed that hair lead levels are significantly and positively associated with MD in the white-matter area adjacent to the right fusiform gyrus [$x, y, z = 46.5, −52.5, −12$, $P = 0.020$, corrected for multiple comparison (threshold-free cluster enhancement (TFCE), permutation), 492.75 mm³, part correlation coefficient for the association between hair pb levels and mean value of the cluster = 0.144] (Fig. 3a, b). Hair lead levels were significantly and negatively associated with MD in extensive areas of the left hemisphere, mainly spreading through the prefrontal cortex, insula, temporal lobe, parietal lobe, putamen, pre- and postcentral gyrus, temporal gyrus, and thalamus (Fig. 3c, d and Table 3). Part correlation coefficients for all the associations between hair lead levels and mean values of significant clusters were <|0.16| and had weak effect sizes.

*Associations of hair lead with brain activity.* Brain activity analyses were performed with data collected from 892 subjects after excluding data containing artifacts or improper behavioral data.

Whole-brain multiple regression analysis showed that head lead levels are significantly and positively associated with the brain activity of the contrast (2-back–0-back) in the pre-supplemental motor area and right premotor cortex (Fig. 4a, b and Table 4). These areas are areas that are activated in the contrast (2-back–0-back). Associations between hair lead levels and mean values of significant clusters showed weak part correlation coefficients.

## Discussion

The present study revealed the associations between lead exposure, as measured by hair lead level, and a wide range of cognitive

measures, brain activity during attention-demanding tasks, and microstructural properties in a large cohort of young adults. Partly consistent with our hypothesis, greater hair lead levels were weakly but significantly associated with increased working memory-related activity in the right premotor and pre-supplemental motor areas. These associations were observed in areas that appeared to be activated during the working memory task. Also partly consistent with our hypothesis, greater hair lead levels were weakly but significantly associated with lower MDDS in the left hemisphere and widespread contingent areas but greater MD in the white-matter area adjacent to the right fusiform gyrus. Partly consistent with a previous study, greater hair lead levels were weakly but significantly associated with lower FA in the right internal capsule and contingent areas. Finally, consistent with previous studies, greater hair lead levels were weakly but significantly associated with lower psychometric intelligence, lower complex arithmetic task scores, lower reading comprehension task scores, and greater impulsivity measures. Not only that, hair lead levels were most robustly, but weakly, associated with novelty seeking and extraversion, which have been associated with dopaminergic functions.

The present findings demonstrate that greater lead levels may be weakly associated with higher brain activity increases in areas showing increased activity depending on the load. Although only two significant clusters were identified in this work, similar patterns were observed in widespread areas showing increased activity (Supplementary Fig. 2). On the one hand, while only micro-level mechanisms may be cited for these neuroimaging observations at present, the results are consistent with some previous findings of the neurotoxicity of lead in nonhuman studies. For example, glutamate levels in the brain have been associated with brain overexcitability[24]. GABA levels in the brain are known to be negatively correlated with brain activation[25]. On the other hand, although lead is known to present many effects, the directions of some of these effects may sometimes be complex.

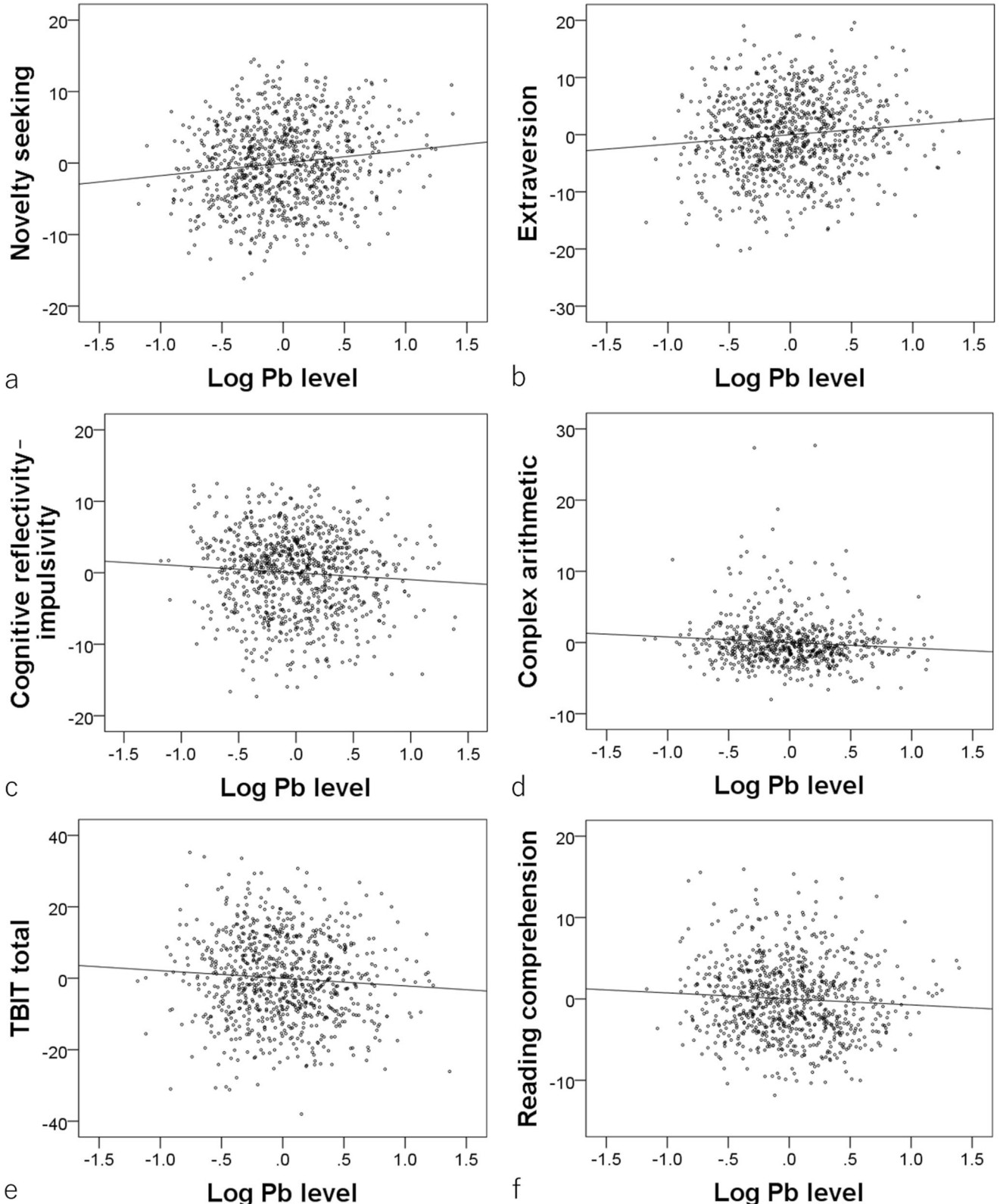

**Fig. 1 Associations between hair lead (Pb) levels and psychological variables.** Partial residual plots with trend lines depicting the associations between the residuals of psychological variables and those of the logarithms of hair lead levels with other confounding factors controlled. Greater hair lead levels (log values) were significantly associated with **a** higher novelty seeking scores ($N = 919$), **b** greater extraversion ($N = 920$), **c** lower cognitive reflexibility–impulsivity scores (i.e., higher impulsivity) ($N = 918$), **d** lower performance on the complex arithmetic task ($N = 661$), **e** lower total intelligence scores on the Tanaka B-type intelligence test (TBIT) ($N = 843$), and **f** lower reading comprehension test scores ($N = 837$).

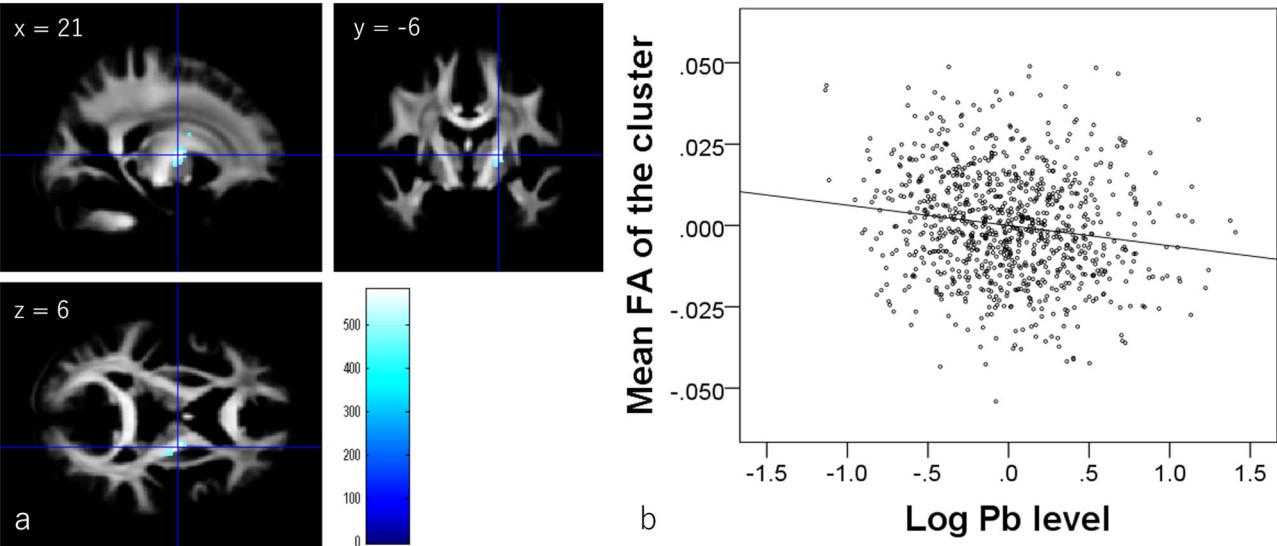

**Fig. 2 Correlation of negative FA with hair lead (Pb) levels ($N = 919$). a** Regions with significant negative correlations between FA and hair lead levels are overlaid on the mean preprocessed (including normalization) but not smoothed, FA images of participants from whom the DARTEL template was created (meaning this mean image is in the normalized space). Results were obtained using a threshold-free cluster enhancement of $p < 0.025$ based on 5000 permutations. Results were corrected at the whole-brain level. The color bar represents the TFCE score. It reflects both voxel's height and the sum of the spatially contiguous voxels supporting it; therefore, it reflects both the strength and extent of effects. Significant correlations were found in areas around the right internal capsule. **b** Scatter plot of the associations between hair lead levels and mean FA in the cluster in (**a**).

Lead accumulation, for instance, is known to reduce glutamine synthetase activity, which, in turn, causes glutamate accumulation; lead also disrupts the GABA pathway and decreases GABA release[9]. These effects on glutamate and GABA may cause brain activation to increase during cognitive tasks. Research has shown that young adults with higher intelligence scores show generally lower working memory-related brain activity increases in areas that are active during a task[26], consistent with neural efficiency theory[27]. This type of inefficient brain activity may partly underscore the observed weak associations between poorer performance in higher-order cognitive tasks and greater lead levels.

Partly consistent with our hypothesis and the findings of previous studies, we found that greater hair lead levels are weakly but significantly associated with lower FA in the white-matter area of the right internal capsule and contingent areas. A previous study revealed that young adults ($N = 159$) with high levels of blood lead show lower FA in widespread white-matter areas, including the internal capsule[16]; a significant weak association between lead level and FA in white-matter areas was also observed in the present study. That this study showed significant weak associations in highly limited areas despite having a greater sample size may be attributed, at least in part, to the use of an inappropriate cluster size test, which has been shown to provide inflated false positive rates[23], in the previous study. The present study supports previous findings by using a large sample size and permutation-based statistics, both of which are known to control for false positives properly[28]. Increased myelination is believed to lead to greater FA in DTI[29]. This FA finding is consistent with the results of a number of nonhuman studies showing that lead delays myelin accumulation[12] and alters the structure of the myelin sheath[13].

Higher hair lead levels were weakly but significantly associated with greater MD in the white-matter area adjacent to the right fusiform gyrus, consistent with previous findings demonstrating that lead damages neural mechanisms in various ways. The areas in which these associations between greater MD and higher hair lead levels were observed did not correspond to areas showing significant associations between greater FA and higher hair lead.

This finding indicates the possible existence of mechanisms through which MD is increased whereas FA is unaffected. For example, lead accumulates in the mitochondria and eventually causes cell apoptosis[10]. Lead accumulation results in reduced glutamine synthetase activity, which, in turn, causes glutamate accumulation and excitotoxicity[10], the effects of which are highly lethal to neurons. Lead is also known to cause oxidative stress via enhancing lipid peroxidation[30,31]. On the other hand, it is known that cell apoptosis in the brain leads to increased MD[32] and that oxidative stress is associated with increase of MD (or apparent diffusion coefficients)[33]. Therefore, these mechanisms may damage neurons or other tissue components to increase MD and slightly reduce higher-order cognitive functions at daily levels of exposure.

Greater hair lead levels were also weakly and significantly associated with lower MDDS and contingent widespread areas of the left hemisphere. The mechanisms behind these associations remain unclear. A previous study investigating the MD of white matter reported similar mixed results of significant positive associations between lead level and MD and significant negative associations between lead level and MD[16]. The effects of lead on the dopaminergic system are complex. Lead is known to prohibit the stimulated release of dopamine but enhance its spontaneous release[9]. Low-level lead exposure only slightly affects dopamine D1 receptors but remarkably affects D2 receptors; moreover, D2 receptors in the striatum decrease whereas D2 receptors in the nucleus accumbens increase[10]. Although the exact mechanisms behind these effects are unclear, increased spontaneous dopaminergic release may lead to decreased MDDS and MD of contingent areas. For example, increases in regular dopamine release and the related neural activity may lead to decreases in MD through activity-induced changes in the swelling of astrocytes, number of synaptic vesicles, and dendritic sprouting, among others[20]. Low dopamine D2 receptor levels and low stimulation of dopamine release have been suggested to indicate greater trait impulsivity and sensation seeking[34]. These alterations in the dopaminergic system may lead to greater impulsivity and upper-level traits (e.g., novelty seeking). However, such suppositions are

**Table 2 Brain regions exhibiting significant negative correlations between hair lead level and fractional anisotropy.**

| No. | Included large bundles[a] (number of significant voxels in the left and right sides of each anatomical area) | x | y | z | TFCE value | Part correlation coefficient[b] | Corrected p value (FWE) | Cluster size (mm³) |
|---|---|---|---|---|---|---|---|---|
| (1) | Posterior limb of the internal capsule (R:49) | 21 | −6 | 6 | 581.6 | −0.154 | 0.008 | 614.25 |
| (2) | Anterior limb of the internal capsule (R:2)/superior corona radiata (R:3)/superior fronto-occipital fasciculus (R:4) | 22.5 | 3 | 19.5 | 449.44 | −0.131 | 0.024 | 20.25 |

[a]Anatomical labels and significant clusters of major white-matter fibers were determined using the ICBM DTI-81 Atlas (http://www.loni.ucla.edu/). [b]Part correlation coefficients of the relationships between hair lead level and mean FA of the significant clusters after controlling for other covariates. The correlation coefficients of significant areas in the whole-brain multiple regression analyses generally do not reflect true effect sizes because of overfitting effects, which are affected by multiple factors, including sample size[95].

speculative; future studies using other experimental methods should be conducted to investigate the exact mechanisms behind the weak associations observed in this work.

It has been shown that whole-brain analyses tend to over-estimate effect sizes, especially in small sample sizes[35] and that small sample studies tend to have greater effect sizes than meta-analyses[36]. In our study, we recruited hundreds of subjects and observed relatively small effects sizes across measures ($|r| < 0.14$). However, due to the reasons described above, despite the relatively large sample size, the true effect size between hair lead levels and psychological and imaging measure may still be smaller than what we reported. Previous meta-analyses also reported small effect sizes ($|r| < 0.16$) between lead levels and individual cognitive differences[2]. However, these could be affected by measurement error when measuring lead and cognitive functions as well as low variances of lead levels among the normal population and not equal to low associations of brain mechanisms with cognitive mechanisms or lead levels.

This study presents some limitations. First, this research is a cross-sectional macro-level neuroimaging study; as such, ultimately, we can neither prove causal relationships nor draw strong conclusions regarding the micro-level mechanisms underlying the observed macro-level neuroimaging and behavioral associations. Second, we used hair lead levels in this study. Previous neuroimaging studies used tibia lead, blood lead level, and lead level of the region in which subjects live instead of individual lead levels measured in the body[14,15,18]. Each method presents inherent weaknesses and strengths. For example, the lead level of the region in which subjects live cannot reflect individual differences in lead intake or exposure, and measurements of tibia lead level are difficult to obtain. Moreover, the blood reflect short-term lead levels, but lead persists in the body for very long periods of time[9]. Hair lead levels can measure the mid-term lead exposure of each individual (i.e., several months) and are generally suitable for long-term measurements[35]. Blood and hair measurements have been suggested to be mutually complementary[37]. However, hair measurement is fragile to external manipulations of hair[4]. Extensive efforts have been made to remove lead from the environment and daily necessities. In addition, in the present study, a history of external hair manipulation showed low correlation (see "Methods"). Empirically, hair and blood lead levels show comparable effect sizes with psychological measures that are supposed to be affected by exposure to lead[4]. These findings show the practical validity of the use of hair lead analysis in modern research. Last, this study is part of a long-standing project with data being collected gradually since 2008. Due to the nature of this study, structural scan protocols could not be updated and have remained fixed since the beginning. While this allowed us to have a good sample size, it limits the use of newer protocols such as neurite orientation dispersion and density imaging, which could have added extra depth to the study. Future studies may need to investigate the effects of lead on brain function with more up-to-date imaging protocols.

In conclusion, we investigated the associations of hair lead levels with cognitive measures, brain activity, FA of the white matter, and MD of gray- and white-matter areas in a large cohort of typically developing young adults. Greater lead levels were weakly but significantly associated with increase of working memory-related activity in the areas of WM network, which show WM-related brain activity increase, lower FA in the right internal capsule and contingent areas, greater MD in the white-matter area close to the right fusiform gyrus, and lower MDDS and widespread contingent areas in gray and white-matter areas of the left hemisphere. Greater hair lead levels also showed weak but significant associations not only with lower psychometric intelligence, lower complex arithmetic task scores, lower reading

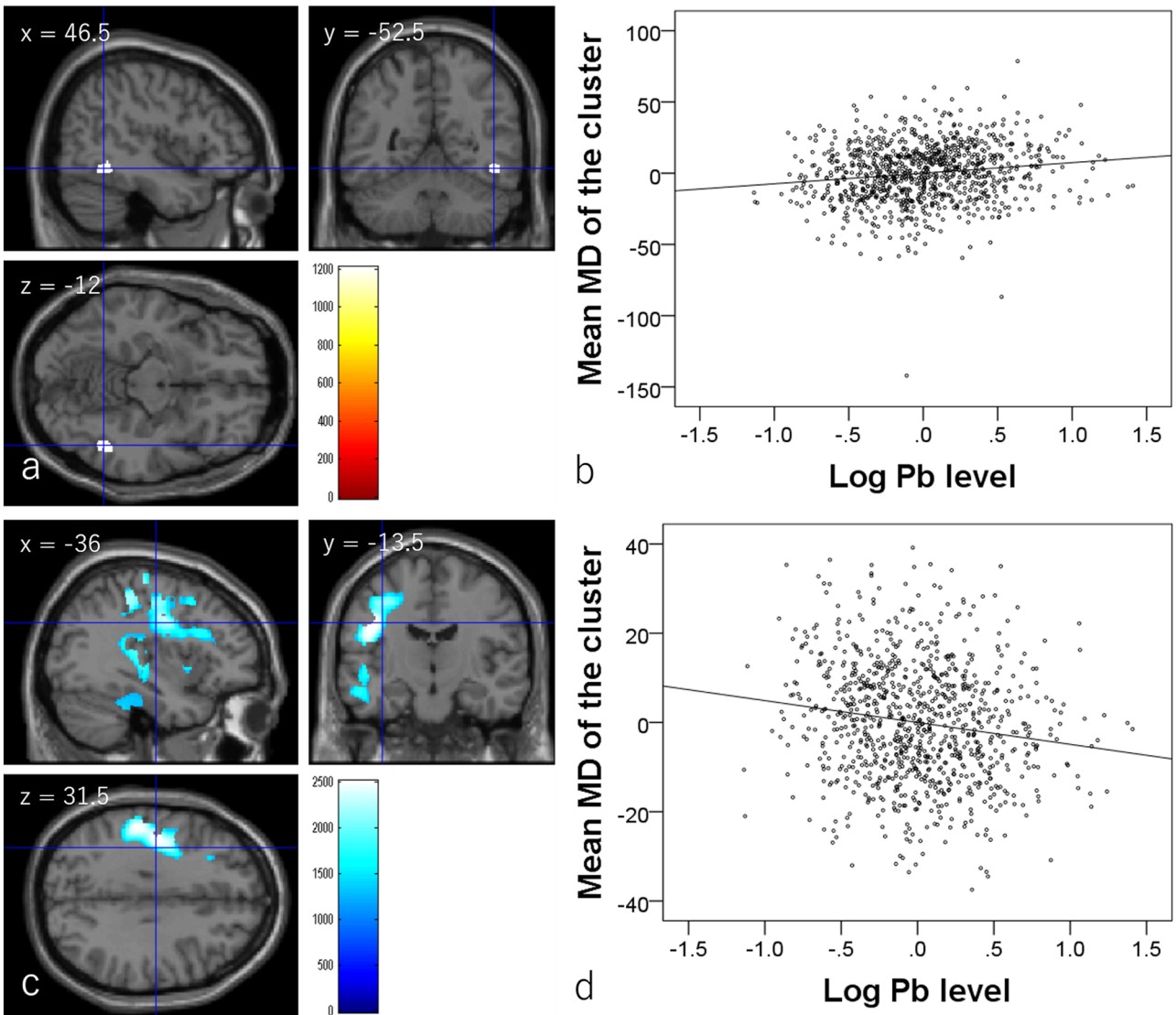

**Fig. 3 Associations between hair lead (Pb) levels and mean diffusivity (MD) (N = 919). a** Regions with significant positive correlations between hair lead levels are overlaid on a single-subject T1 image from SPM8 (the image file with the name of "single_subje_T1"). Results were obtained using a threshold-free cluster enhancement (TFCE) of $p < 0.025$ based on 5000 permutations. Results were corrected at the whole-brain level. The color bar represents the TFCE score. It reflects both voxel's height and the sum of the spatially contiguous voxels supporting it; therefore, it reflects both the strength and extent of effects. Significant correlations were found in white-matter areas adjacent to the right fusiform gyrus. **b** Scatter plot of the associations between hair lead levels and mean MD in the cluster in (**a**). **c** Regions with significant negative correlations between hair lead levels and MD are overlaid on a single-subject T1 image from SPM8 (the image file with the name of "single_subje_T1"). Results were obtained using a TFCE of $p < 0.025$ based on 5000 permutations. Results were corrected at the whole-brain level. The color bar represents the TFCE score. It reflects both voxel's height and the sum of the spatially contiguous voxels supporting it; therefore, it reflects both the strength and extent of effects. Significant correlations were found in extensive gray and white-matter areas of the left hemisphere. **d** Scatter plot of the associations between hair lead levels and mean MD in the largest cluster in (**c**).

comprehension task scores, greater impulsivity measures, but also with and traits that had been associated with the dopaminergic system, such as greater novelty seeking and extraversion. These findings reflect the weak effects of daily lead level on the excitability and microstructural properties of brains, particularly in the dopaminergic system.

## Methods

**Subjects**. The present study is part of an ongoing project that aims to investigate associations between brain imaging, cognitive functions, and aging and included 920 healthy right-handed individuals (561 men and 359 women) from whom the data necessary for whole-brain analyses involving lead levels were collected. The mean subject age was 20.7 years (standard deviation, 1.8; age range: 18–27 years). Written informed consent was obtained from adult subjects or the parents (guardians) of nonadult subjects (age <20 years). This study was approved by the

Ethics Committee of Tohoku University. All procedures performed in studies involving human participants were in accordance with the ethical standards of the institutional and/or national research committee and with the 1964 Helsinki declaration and its later amendments or comparable ethical standards.

Some subjects who participated in this study also participated in our intervention studies (psychological data and imaging data recorded before the intervention were used in the present study)[36]. Psychological tests and MRI scans not described here were performed together with those described in the present study. All subjects were either undergraduate students, graduate students, or fresh graduates. All subjects had normal vision and none had neurological or psychiatric illnesses. Handedness was evaluated using the Edinburgh Handedness Inventory[38].

Subjects were instructed to get sufficient sleep, maintain their condition, eat sufficient breakfast, and consume their usual amount of caffeinated foods/drinks on the day of cognitive testing and MRI scans. Subjects were also instructed to avoid alcohol the night before the assessment.

The description in this subsection is mostly reproduced from our previous study of the same project[39].

**Table 3 Brain regions exhibiting significant negative correlations between hair lead level and mean diffusivity.**

| No. | Included gray-matter areas[a] (number of significant voxels in the left and right sides of each anatomical area) | Included large bundles[b] (number of significant voxels in the left and right sides of each anatomical area) | x | y | z | TFCE value | Part correlation coefficient | Corrected p value (FWE) | Cluster size (mm³) |
|---|---|---|---|---|---|---|---|---|---|
| (1) | Angular gyrus (L:15)/inferior frontal operculum (L:180)/inferior frontal triangular (L:228)/middle frontal other areas (L:998)/superior frontal other areas (L:471)/Heschl gyrus (L:89)/hippocampus (L:87)/insula (L:481)/paracentral lobule (L:187)/inferior parietal lobule (L:467)/superior parietal lobule (L:132)/postcentral gyrus (L:1979)/precentral gyrus (L:2279)/precuneus (L:9)/Putamen (L:181)/Rolandic operculum (L:442)/supplemental motor area (L:6)/supramarginal gyrus (L:888)/inferior temporal gyrus (L:87)/middle temporal gyrus (L:1064)/superior temporal gyrus (L:836)/thalamus (L:230) | Posterior limb of the internal capsule (L:43)/retrolenticular part of the internal capsule (L:548)/anterior corona radiata (L:29)/superior corona radiata (L:43)/posterior thalamic radiation (L:9)/sagittal stratum (L:103)/external capsule (L:383)/Stria terminalis (L:112)/superior longitudinal fasciculus (L:805)/inferior fronto-occipital fasciculus (L:108) | −42 | −13.5 | 31.5 | 2509.93 | −0.150 | 0.002 | 51036.75 |
| (2) | Fusiform gyrus (L:465)/parahippocampal gyrus (L:20)/inferior temporal gyrus (L:13)/cerebellum (L:219) | None | −33 | −39 | −30 | 1382.16 | −0.128 | 0.016 | 2338.875 |
| (3) | Inferior temporal gyrus (L:47) | None | −49.50 | −36 | −7.5 | 1249.67 | −0.091 | 0.023 | 40.5 |
| (4) | Middle temporal gyrus (L:3) | None | −48 | −33 | −25.5 | 1242.95 | −0.118 | 0.023 | 158.625 |

[a]The labels of the anatomical regions of gray matter were based on the WFU PickAtlas Tool (http://www.fmri.wfubmc.edu/cms/software#PickAtlas/)[96,97] and the PickAtlas automated anatomical labeling Atlas option[98]. Temporal pole areas included all subregions in the areas of this Atlas.
[b]The anatomical labels and significant clusters of major white-matter fibers were determined using the ICBM DTI-81 Atlas (http://www.loni.ucla.edu/).

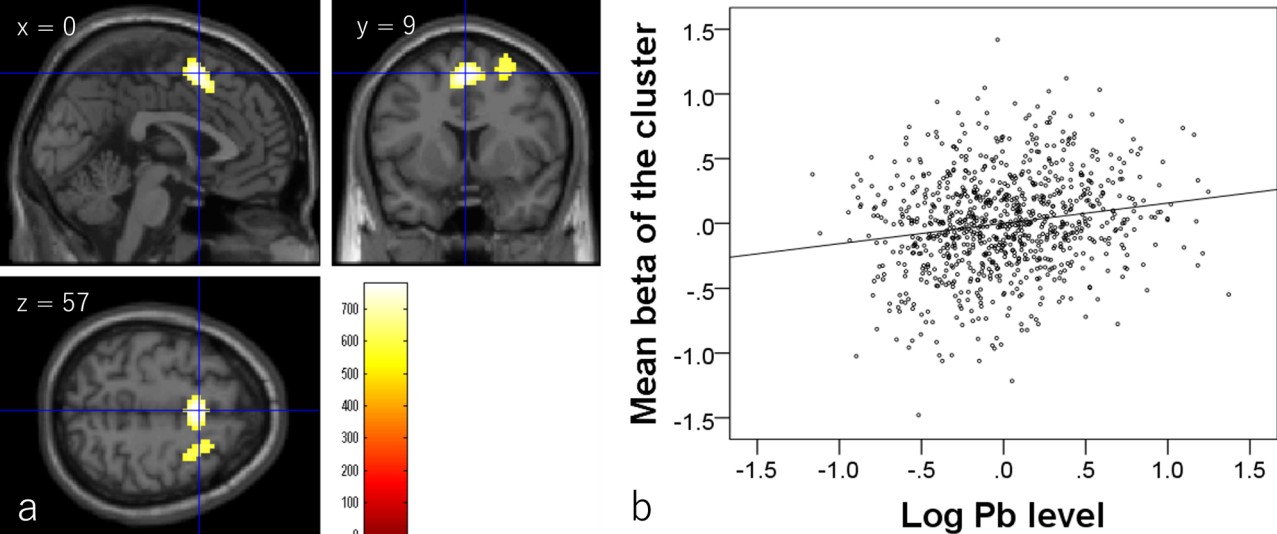

**Fig. 4 Correlation of brain activity with hair lead (Pb) levels ($N = 892$). a** Regions with significant correlations between the brain activity of the contrast (2-back–0-back) and hair lead levels are overlaid on a single-subject T1 image from SPM8 (the image file with the name of "single_subje_T1"). Results were obtained using a threshold-free cluster enhancement of $p < 0.025$ based on 5000 permutations. The color bar represents the TFCE score. It reflects both voxel's height and the sum of the spatially contiguous voxels supporting it; therefore, it reflects both the strength and extent of effects. Significant positive correlations were found in the presupplementary motor area and right middle and superior frontal gyrus. **b** Scatter plot of the association between hair lead levels and mean beta estimates in the larger clusters in (**a**).

**Informed consent**. Informed consent was obtained from all individual participants included in the study.

**Details of recruitment and exclusion criteria of subjects**. They were recruited using advertisements on bulletin boards at Tohoku University or via email introducing the study. These advertisements and emails specified the unacceptable conditions in individuals with regard to participation in the study such as handedness, the existence of metal in and around the body, claustrophobia, the use of certain drugs, a history of certain psychiatric and neurological diseases, and previous participation in related experiments.

A history of psychiatric and neurological diseases and/or recent drug use was assessed using our laboratory's routine questionnaire, in which each subject answered questions related to their current or previous experiences of any of the listed diseases and listed drugs that they had recently taken. Drug screening was performed to confirm that the subjects were not taking any illegal psychostimulants or antipsychotic drugs, which was one of the exclusion criteria used during the course of the recruitment. Subjects with exclusion criteria should have been excluded before they came to the lab, but if they came for some reason, they had to go back once it was found that they met an exclusion criterion. Consequently, none had a history of neurological or psychiatric illness. In the course of this experiment, the scans were checked for obvious brain lesions and tumors, but there were no subjects having such obvious lesions or tumors.

These descriptions are mostly obtained from our previously published work[40].

**Hair acquisition and hair mineral analysis**. Hair is a repository of all elements that enter the body, and mineral levels in hair reflect accumulation of mineral composition over several months to years[41]. And thus, hair mineral levels are not affected substantially by rapid fluctuation in mineral intake and show long-term stability[35]. These characteristics give hair mineral analysis advantages over other methods to measure mineral levels such as blood and urine analyses. Studies showed positive correlation between concentrations of basic elements in the hair and in the body[42,43]. However, it was also suggested that hair mineral analysis requires, sampling by trained personnel, with standardized pre-analytical and analytical procedures, using suitable and sensitive equipment were required to obtain comparable results[44]. And new analytic methods and good practice have improved the precision of hair mineral analysis[45].

Scalp hair samples (~4 cm in length and 0.1 g in weight) were collected from each subject and cut as close to the scalp as possible. Hair samples were sent to the La Belle Vie research laboratory and analyzed using established methods as described previously and reproduced below[46].

A 75 mg hair sample was weighed in a 50 ml plastic tube and then washed twice with acetone and once with 0.01% Triton solution, as recommended by the Hair Analysis Standardization Board[47]. The washed hair sample was mixed with 10 ml 6.25% tetramethylammonium hydroxide (Tama Chemical) and 50 μl 0.1% gold solution (SPEX Certi Prep.) and dissolved at 75 °C with shaking for 2 h. After cooling the solution to room temperature, the internal standard (Sc, Ga, and In)

solution was added, and adjusting the volume gravimetrically, the solution was used for mineral analysis. Mineral concentrations were measured by inductively coupled plasma mass spectrometry (Agilent-7500ce) by the internal standard method[48–50] and are expressed as ng/g hair (ppb). For quality control, we used human hair certified reference materials from the National Institute for Environmental Studies of Japan (NIES CRM no. 13)[51].

The logarithms of mineral levels in hair were analyzed for all measures used because logarithms of hair lead levels were closer to the normal distribution and could alleviate the effects of outliers. This procedure is consistent with previous studies, including those from researchers affiliated with the institution in which the mineral levels of our hair samples were measured (i.e., Research Laboratory, La Belle Vie Inc.) e.g., ref.[52]. The description in this subsection is mostly reproduced from our previous study on the same subject[39].

We also obtained information on when the majority of the participants ($N = 835$) last underwent hair coloring, perming, or bleaching. The possible answers were (a) within 1 month, (b) within 1–2 months, (c) within 2–3 months, and (d) within 3–6 months, and (e) not performed within 6 months. As the hair samples (~4 cm long) were cut as close to the scalp as possible, based on hair growth, the answers were coded as follows: (a) = 3, (b) = 2, (c) = 1, (d) = 0.5, and (e) = 0. Although these values showed significant correlations with hair lead level, the effect size was low ($r = 0.82$–0.12) and analyses that included history of coloring, perming, and bleaching as covariates did not substantially affect the effects size of the significant associations in the present study.

**Psychological measures**. Following neuropsychological testing, several questionnaires were administered to the participants. These tests were chosen because of the known effects of lead on a wide range of cognitive functions, dopaminergic mechanisms, attention deficit, and several mood states, as described earlier. The test descriptions in this subsection are largely reproduced from our previous studies[53].

[A] Raven's advanced progressive matrices[54] is a nonverbal reasoning task and representative measure of general intelligence. More details of this task are available in our previous study[55].

[B] The Tanaka B-type intelligence test (TBIT)[56] type 3B (TBIT) is a nonverbal mass intelligence test used for third-year junior high school and older examinees in Japan. Rather than using story problems, the test uses figures, single numbers, and letters as stimuli. The subjects must solve as many problems as possible within a certain time (a few minutes) in all subtests, which means that these problems are complex cognitive speed tasks. More details on the TBIT are available in our previous study[57].

[C] Two arithmetic tasks measured performance in two forms of one-digit times one-digit multiplication problems (i.e., a simple arithmetic task with numbers between 2 and 9) and two forms of two-digit times two-digit multiplication problems (i.e., a more complex arithmetic task with numbers between 11 and 19). The subjects were asked to solve as many questions as possible in simple and complex arithmetic tasks within 30 and 60 s, respectively.

**Table 4 Brain regions exhibiting significant positive correlations between hair lead level and brain activity.**

| No. | Included gray-matter areas[a] (number of significant voxels in the left and right sides of each anatomical area) | x | y | z | TFCE value | Corrected p value (FWE) | Part correlation coefficient | Cluster size (mm³) | Activated areas, deactivated areas in the 2-back-0-back contrast[b] |
|---|---|---|---|---|---|---|---|---|---|
| 1 | Middle cingulum (R:6)/superior frontal medial area (R:3)/supplemental motor area (L:100, R:72) | 0 | 9 | 57 | 775.37 | 0.006 | 0.169 | 4671 | 100%, 0% |
| 2 | Middle frontal other areas (R:25)/superior frontal other areas (R:61) | 33 | 6 | 33 | 625.11 | 0.018 | 0.163 | 2187 | 100%, 0% |

[a]The labels of the anatomical regions of gray matter were based on the WFU PickAtlas Tool (http://www.fmri.wfubmc.edu/cms/software#PickAtlas/)[96,97] and the PickAtlas automated anatomical labeling Atlas option[98]. Temporal pole areas and some other areas included all subregions in the areas of this Atlas.
[b]Percentage of voxels activated or deactivated in the contrast (2-back-0-back) in our previous study at a threshold of $P < 0.05$, corrected for FDR[26].

[D] Hakoda's version of the Stroop task[58] was used to measure response inhibition and impulsivity. This version of the matching-type Stroop task requires subjects to check whether their chosen answers are correct, unlike the traditional oral-naming Stroop task. The test consists of two control tasks, namely, Word-Color and Color-Word tasks, a Stroop task, and a reverse Stroop task. Reverse Stroop and Stroop interference rates were calculated from the scores obtained from these tasks. Details of this test are provided in our previous study[59].

[E] The reading comprehension task used in this study was developed by Kondo et al.[60]. More details on this test, such as how it was developed and its validity, are provided by Kondo et al.[60] and our previous study[61].

[F] S-A creativity test. Creativity as divergent thinking was measured using the S-A creativity test[62]. More details are available in our previous study[55].

[G] A (computerized) digit-span task, which is a working memory task for details, see ref. [63].

[H] The motivational state of the day for each subject was measured using the Vigor subscale of a shortened Japanese version[64] of the Profile of Mood States psychological rating scale[65].

[I] The Japanese version[66] of the Temperament Character Inventory[67] was used to measure novelty seeking. A subscale of this measure, that is, impulsiveness, was also used to measure impulsivity.

[J] The Japanese version of the NEO Five-Factor Inventory was used to measure extraversion[68].

[K] The cognitive reflectivity–impulsiveness questionnaire[69] was used to assess individual differences in reflectivity and impulsivity[70].

[L] The External-Preoccupation Scale[71] was used to measure the maintenance of external focus on a specific object. Data for this scale were collected only from a subset of the subjects (i.e., 678 successfully genotyped subjects).

[M] The Japanese version[72] of the Beck Depression Inventory[73] was used to measure the current state of depression.

**fMRI task**. fMRI was used to map brain activity during the cognitive tasks. The descriptions of this task are mostly reproduced from a previous study using the same methods[74]. Briefly, the n-back task is a typical fMRI task with conditions of 0-back (simple cognitive process) and 2-back (working memory). Subjects were instructed to judge whether a stimulus, that is, one of four Japanese vowels presented visually, appearing "n" positions earlier is identical to the current stimulus by pushing a button. In the 0-back task, subjects were instructed to determine whether a presented letter is identical to the target stimulus by pushing a button. We used a simple block design. More details on this task are described below.

Participants received instructions for the tasks and practiced the tasks before entering the MRI scanner. During scanning, they viewed stimuli on a screen via a mirror mounted on a head coil. Visual stimuli were presented using Presentation Software (Neurobehavioral Systems, Inc., Albany, CA, USA). A fiber-optic light-sensitive key press interface with a button box was used to record participants' task responses.

Two conditions were used: 0-back and 2-back. Each condition had six blocks, and all N-back tasks were performed in one session. Subjects were instructed to recall visually presented stimuli (four Japanese vowels) presented "n" stimuli before the currently presented stimulus (e.g., participants had to recall the letter presented two letters earlier for the 2-back task or the currently presented letter for the 0-back task). Two buttons were used during the 0-back task: subjects were instructed to push the first button when the defined target stimuli were presented and the second button when nontarget stimuli were presented. During the 2-back task, subjects were instructed to push the first button when the currently presented stimulus and the stimulus presented two stimuli earlier were the same, and to push the second button when the currently presented stimulus and the stimulus presented two stimuli earlier differed. Since the four stimuli were presented randomly, the ratio of matched trials to unmatched trials was 1:3 on average. Our version of the N-back task was designed to require individuals to push buttons continuously during the task period. The task level of the memory load was presented above the stimuli for 2 s before the task started and remained visible and unchanged during the task period (cue phase). Each letter stimulus was presented for 0.5 s with a fixation cross presented for 1.5 s between items. Each block consisted of ten stimuli. Thus, each block lasted 20 s. A baseline fixation cross was presented for 13 s between the last task item and the presentation of the next task level of the memory load (start of the cue phase). Thus, the rest period lasted for 15 s (13 s + 2 s). There were six blocks for each 2- and 0-back condition. The descriptions in this subsection were mostly reproduced from another study of ours from the same project using the same methods[74].

**Consideration and exclusion of movement effects during fMRI analyses**. Thorough instructions and thorough fixation by the pad were given as much as possible to prevent head motion during the fMRI scan. We did not exclude any subject from the fMRI analyses based on excessive motion during the scan. However, we excluded subjects if artifacts were visually apparent on their scans, regardless of the cause of these artifacts. The subjects were young adults and the scan did not last for long. Only six subjects' maximum movement from the original point in one of the directions exceeded 3 mm, and removing these subjects from analyses did not substantially alter the significant results of the present study. Furthermore, frame-wise displacement during fMRI scan did not significantly

correlate with hair lead levels after all the other covariates of the whole-brain analyses of $n$-back tasks were controlled (partial correlation analysis, partial correlation coefficient = −0.008, $p = 0.801$). These procedures are identical to those described in our previous study[39], and the descriptions in this subsection are mostly reproduced from that study.

*Image acquisition.* The MRI acquisition methods are described in our previous study and reproduced below[75]. All of the MRI data were acquired using a 3T Philips Achieva scanner. Diffusion-weighted data were acquired using a spin-echo EPI sequence (TR = 10293 ms, TE = 55 ms, FOV = 22.4 cm, $2 \times 2 \times 2$ mm$^3$ voxels, 60 slices, SENSE reduction factor = 2, number of acquisitions = 1). The diffusion weighting was isotropically distributed along 32 directions ($b$ value = 1000 s/mm$^2$). In addition, three images with no diffusion weighting ($b$ value = 0 s/mm$^2$) ($b = 0$ images) were acquired using a spin-echo EPI sequence (TR = 10,293 ms, TE = 55 ms, FOV = 22.4 cm, $2 \times 2 \times 2$ mm$^3$ voxels, 60 slices). FA and MD maps were calculated from the images collected using a commercially available diffusion tensor analysis package on the MR console. The descriptions in this subsection are mostly reproduced from a previous study using similar methods[76]. The acquisitions for phase correction and signal stabilization were not used as reconstructed images. MD and FA maps were calculated from the collected images using a commercially available diffusion tensor analysis package on the MR console. This method has been used in many of our studies[77–81]. The image-generated results are congruent with those of previous studies using other methods[82,83], suggesting the validity of this method. The procedures involved correction for motion and distortion caused by eddy currents. Calculations were performed according to a previously proposed method[84].

Forty-two transaxial gradient-echo images (TR = 2.5 s, TE = 30 ms, flip angle = 90°, slice thickness = 3 mm, FOV = 192 mm, matrix = $64 \times 64$) covering the entire brain were acquired using an echo planar sequence. A total of 174 functional volumes were obtained for the $n$-back sessions.

**Preprocessing of structural data.** Preprocessing and analysis of functional activation data were performed using SPM8 implemented in MATLAB. Descriptions in this subsection were mostly reproduced from a previous study using similar methods[74]. Before analysis, individual BOLD images were realigned and resliced to the mean BOLD image, and corrected for slice timing. The mean BOLD image was then realigned to the mean $b = 0$ image together with the slice-timing-corrected images, as described previously[63]. Because the mean $b = 0$ image was aligned with the FA image and MD map, the BOLD image, $b = 0$ image, FA image, and MD map were all aligned. Subsequently, using a previously validated two-step segmentation algorithm of diffusion images and diffeomorphic anatomical registration through an exponentiated lie algebra (DARTEL)-based registration process[78], all images—including gray-matter segments [regional gray-matter density (rGMD) map], white-matter segments [regional white-matter density (rWMD) map], and cerebrospinal fluid (CSF) segments [regional CSF density map] of the diffusion images—were normalized.

The details of these procedures, which were also described in our previous study[78], are as follows. Using the new segmentation algorithm implemented in SPM8, FA images of each individual were segmented into six tissues (first new segmentation). The default parameters and tissue probability maps were used in this process, except that affine regularization was performed using the International Consortium for Brain Mapping template for East Asian brains and the sampling distance (approximate distance between sampled points when estimating the model parameters) was 2 mm. We then synthesized the FA image and MD map. In the synthesized image, the area with a WM tissue probability >0.5 in the abovementioned new segmentation process was the FA image multiplied by −1 (hence, the synthesized image shows very clear contrast between WM and other tissues); the remaining area is the MD map (for details of this procedure, see below). The synthesized image from each individual was then segmented using the new segmentation algorithm implemented in SPM8 with the same parameters as above (second new segmentation). This two-step segmentation process was adopted because the FA image has a relatively clear contrast between GM and WM, as well as between WM and CSF, and the first new segmentation step can segment WM from other tissues. On the other hand, the MD map has clear contrast between GM and CSF and the second new segmentation can segment GM. Since the MD map alone lacks clear contrast between WM and GM, we must use a synthesized image (and the two-step segmentation process).

We then proceeded to the DARTEL registration process implemented in SPM8. We used the DARTEL import image of the GM tissue probability map produced in the second new segmentation process as the GM input for the DARTEL process. The WM input for the DARTEL process was created as follows. First, the raw FA image was multiplied by the WM tissue probability map from the second new segmentation process within the areas with a WM probability >0.5 (signals from other areas were set to 0). Next, the FA image * WM tissue probability map was coregistered and resliced to the DARTEL import WM tissue probability image from the second segmentation. The template for the DARTEL procedures was created using imaging data from 63 subjects who participated in the experiment in our lab[63] and were included in the present study (meaning that they have the same characteristics as the subjects in this study). The first reason why we created the

DARTEL template from the images of a subset of all subjects (63 subjects) and not from all subjects is because this is a large sample for creating a template compared to previous studies and thus cannot be considered problematic. The second reason is that the project in which the subjects participated is ongoing, and the DARTEL processes—especially our processes—require vast amounts of time and the resultant images require large storage resources; thus, we cannot reprocess the images of all subjects and add newer images whenever we change the number of subjects. Next, using this existing template, the DARTEL procedures were performed for all subjects in this study. In these procedures, the parameters were changed as follows to improve accuracy. The number of Gauss–Newton iterations performed within each outer iteration was set to 10 and, in each outer iteration, we used eightfold more timepoints to solve the partial differential equations than the default values. The number of cycles used by the full multigrid matrix solver was set to 8. The number of relaxation iterations performed in each multigrid cycle was also set to 8. The resultant synthesized images were spatially normalized to MNI space. Using these parameters, the raw FA map, rGMD, and rWMD map from the abovementioned second new segmentation process were normalized to give images with $1.5 \times 1.5 \times 1.5$ mm$^3$ voxels. The FA image * WM tissue probability map was used in the DARTEL procedures because it includes different signal intensities within WM tissues and the normalization procedure can take advantage of intensity differences to adjust the image to the template from the perspective of the outer edge of the tissue and within the WM tissue. No modulation was performed in the normalization procedure.

The voxel size of the normalized FA, MD, and segmented images was $1.5 \times 1.5 \times 1.5$ mm$^3$. The voxel size of the normalized BOLD images was $3 \times 3 \times 3$ mm$^3$.

Next, we created average images of normalized rGMD and rWMD images from the normalized rGMD and rWMD images from the subset of the entire sample (63 subjects)[78]. From the average image of normalized WM segmentation images from the 63 subjects mentioned above, we created mask image consisting of voxels with a WM signal intensity >0.99. We then applied this mask image to the normalized FA image, thereby only retaining areas highly likely to be white matter. These images were smoothed (6 mm full-width half-maximum) and carried through to the second-level analyses of FA. As described previously[78], through application of the mask, images unlikely to be WM or border areas between WM and other tissues were removed. The FA images were not affected by signals from tissues other than WM even after smoothing. This is important considering that, in these areas, WM volume and FA are highly correlated[85] and the FA map supposedly reflects the extent of WM. Further, differences in WMC compared with other tissues among individuals can be ignored after application of this mask because, within the masks, all voxels show very high white-mater probability. For validation of these preprocessing methods and comparison with other methods, see the supplementary online material of our previous study[78].

Through these procedures, we believe that we successfully mitigated or removed the problems of voxel-based analysis of FA analysis raised by Smith et al[86]. These problems include (a) misalignment within white-matter tissue (addressed by new segmentation processes and DARTEL processes that utilized difference in signal distribution within white-matter using the FA signal) and (b) the effects of different tissue types and partial volume effects (addressed by new segmentation processes, the DARTEL processes, and application of the mask confined to images highly likely to be white matter (in the case of MD maps, white matter or gray matter)). Through these methods, the white matter of DTI images as well as the gray-matter areas of DTI images become available for analysis. We avoided co-registration of DTI images to T1-weighted structural images because the shapes differ due to the unignorable distortion of EPI images in 3T MRI.

**Statistics and reproducibility**

*First-level analysis of functional imaging data.* The following descriptions are mostly reproduced from our previous study using similar methods[74]. Individual-level statistical analyses were performed using a general linear model. A design matrix was fitted to each participant with one regressor in each task condition (0- or 2-back in the $n$-back task) by using the standard hemodynamic response function. The cue phases of the $n$-back task were modeled in the same manner but not analyzed further. Six parameters obtained by rigid body corrections for head motion were regressed out by adding these variances to the regressor. The design matrix weighted each raw image according to its overall variability to reduce the impact of movement artifacts[87]. We removed low-frequency fluctuations using a high-pass filter with a cut-off value of 128 s. After estimation, beta images of contrasts of (2-back > 0-back) were smoothed (8 mm FWHM) and used for second-level analyses.

*Statistical analyses of non-whole-brain analyses.* Behavioral data were analyzed using R software version 4.0.1[88], and the associations of hair lead levels with psychological outcome measures were tested using multiple regression analyses. A total of 16 cognitive variables were included as dependent, as presented in Table 1. The independent variables included sex, age, self-reported height, self-reported weight, body mass index (calculated from self-reported height and weight), annual family income, parents' highest educational qualifications (measured as reported in ref. [89]), and hair lead levels. $P$ values were assessed by permutation (5000 iterations) based on multiple regression analyses using the ImPerm package[90] and R software.

The expression for each test is as follows.

Result_x < − lmp(Test_x ∼ sex + age + height + weight + BMI + parents_education_level + family_income + hair_lead_level, datasetname_y, seqs = TRUE)

Summary(Result_x)

Permutation analyses were conducted nine times (conducting nine times lead to more stable results and the number was chosen empirically) for each cognitive measure, and the median $p$ value was used for analyses. For all analyses, results with a threshold of $p < 0.05$ (two-sided) after correcting for the false discovery rate using a two-stage sharpened method[91] were considered statistically significant.

*Whole-brain statistical analysis.* We investigated whether the imaging measures are associated with individual differences in hair lead. Whole-brain multiple regression analyses were performed using SPM8. All the data necessary for diffusion data were properly obtained from 919 subjects and all the data necessary for functional data were properly obtained from 891 subjects and analyses were performed with those data.

The covariates used in the FA and MD analyses were identical to those applied for psychological analyses; volume-level mean frame-wise displacement during the diffusion scan was also added as a covariate for these analyses. FA analysis was performed within the white-matter mask created above, and MD analysis was performed within the gray-matter + white-matter mask.

In the fMRI analyses, the maps of dependent variables were beta estimate images of 2-back > 0-back contrast. The covariates used for this analysis included those used in the psychological analyses, as well as accuracies and reaction times in the 0-back and 2-back tasks and volume-level mean frame-wise displacement during the scan for the *n*-back task[92].

Correction for multiple comparisons was performed using TFCE[93] with randomized (5000 permutations) nonparametric testing using the TFCE toolbox (http://dbm.neuro.uni-jena.de/tfce/). The family-wise error threshold was corrected at $p < 0.025$ (one-tailed).

*Rationale for the use of SPM8 in the preprocessing of DTI/fMRI data and statistical analyses.* Concerning preprocessing, we made use of SPM8 because our procedure is unique and has previously been validated with SPM8[78]. Furthermore, when we use SPM12 and the same parameter sets that were validated in SPM8, apparent misclassifications of tissue types in certain brain areas repeatedly occur during the segmentation processes.

As for the 1st level analysis, we kept using SPM8's robust WLS's 1st level model specification and estimation, because result maps of SPM8's robust WLS's 1st level model specification and SPM8's robust WLS's estimation and those of SPM12's robust WLS's 1st level model specification and SPM12's robust WLS's estimation, are identical under default parameter settings. Further, for practical reasons, inconsistency of SPM version in the individual preprocessing and analytical procedures are not preferred in the review process.

As for the second-level analysis, the distribution of *t* values from our results is identical regardless of SPM versions. As long as permutation procedures are taken, the results should not vary between SPM versions. Furthermore, our in-house script is only compatible with SPM8.

**Reporting summary.** Further information on research design is available in the Nature Research Reporting Summary linked to this article.

## Data availability

All the experimental data obtained in the experiment of this study will be available to ones that were admitted in the ethics committee of Tohoku University, School of Medicine. All data sharing activities must be first approved by the Ethics Committee of Tohoku University's medical faculty. The corresponding author is responsible for replying to this request and cooperating. Supplementary Data 1 includes all the independent and dependent variables that were used to generate the residual plots of Figs. 1–4.

## Code availability

The names, versions, and parameters of the software that were used, were provided in "Methods."

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

## Acknowledgements

We respectfully thank Yuki Yamada for operating the MRI scanner, and Haruka Nouchi for being an examiner of psychological tests. We also thank study participants, the other examiners of psychological tests, and all of our colleagues in Institute of Development, Aging and Cancer and in Tohoku University for their support. This study was supported by a Grant-in-Aid for Young Scientists (B) (KAKENHI 23700306) and a Grant-in-Aid for Young Scientists (A) (KAKENHI 25700012) from the Ministry of Education, Culture, Sports, Science, and Technology. The authors would like to thank Enago (www.enago.jp) for the English language review. We would like to thank La Belle Vie Inc. and its employees for the hair mineral level analyses as well as Dr Yasuda and Dr Sonobe for their technical advice regarding the analyses.

## Author contributions

H.T., Y.T., and R.K. designed the study. H.T., R.N., R.Y., Y.K., S.N., A.S., K.I., S.H., T.A., C.M.M., K.S., T.N., S.I., S.Y., M.D., and Y.S. collected the data. H.T. analyzed the data and prepared the manuscript. All authors reviewed the manuscript.

## Competing interests

The authors declare no competing interests.
