## [Peer Review File · Communications Biology]

Reviewers' comments:

Reviewer #1 (Remarks to the Author):

In their article, "Associations of lead exposure with functional and microstructural neural mechanisms", the authors explore the effect of lead on the brain using diffusion MRI and fMRI with a battery of various cognitive tasks results.

Thanks to a large dataset, this study shows the association between hair lead levels and changes in healthy young adults' structural and functional activity and cognitive measures. The article shows promising results to study the impact of lead exposure further. I have a few remarks and modifications that I think would help the reader better understand the methods and results. I also want to note that I appreciate the prudence of the discussion regarding the results' causality. I think this is really important.

Major remarks:

Methods-Preprocessing:

Why did the authors use SPM8 to preprocess the data? This version is vastly outdated and requires using an obsolete version of Matlab as well. It exposes the analyses to potential bugs and much less accurate methods that nowadays standards. I think the authors should acknowledge this in the limitations.

Statistical analyses of non-whole-brain analyses:

"Permutation analyses were conducted nine times for each cognitive measure, and the median p-value was used for analyses." Why nine times? Is there any explanation for that number?

Discussion:

"Lead accumulation results in reduced glutamine synthetase activity, which, in turn, causes glutamate accumulation and excitotoxicity, the effects of which are highly lethal to neurons. Lead is also known to cause oxidative stress via enhancing lipid peroxidation. These mechanisms may damage neurons or other tissue components to increase MD and reduce higher-order cognitive functions." This part gives a lot of hypotheses but could the authors provide some references to back them up? Also, even the rest of the paragraph only contains one reference, it would be helpful to have these mechanisms shown by other studies.

Figures:

Why figure 3 is overlaid on the single-patient image from SPM when the others are in the DARTEL space? DARTEL is the template used to register all the images, no?

Minor remarks:

Statistical analyses of non-whole-brain analyses:

Please provide a more detailed description of the parameters. I reckon the ImPerm package offers multiple tools, and it would be easier for the readers to reproduce the analyses.

Reviewer #2 (Remarks to the Author):

Dear editor,

Thank you for inviting me to review this article. I read the work with interest. The article investigates the functional microstructural correlates of lead exposure.

While the principle aim behind this article is sound, I have certain concerns that I will elucidate in the following lists.

Major concerns:

1) It is commendable that the authors produce regression graphs in their figure. Unfortunately, the effects shown in figure one are very small. Therefore the authors should ensure that any claims that are made in the article are proportionate to the small effect sizes found in the analysis. Further, the authors should explicitly give measurements of effect sizes in the text and explicitly comment on them.

2) It is disappointing that the authors have chosen to use metrics such as FA, MD and other tensor scalars when much more advanced (and more biologically interpretable metrics are available - see NODDI, CHARMED, CSD fiber density etc ...). Further, it seems (unless I have misunderstood) that this work reuses a dataset from Takeuchi (2012). This is over 10 years ago and the quality of data available now is superior (32 directions at a b value of 1000 s/mm² is rather low). I would like to know why this dataset was chosen. This choice should be made explicitly in the manuscript. If on the other hand, this is not a reuse of data but a reuse of protocol, then I do not understand why the protocol was not updated. Perhaps I have misunderstood.

3) I do not understand why the software used for analysis (SPM) is not the most recent version, which has been available for many years now.

Minor concerns:

1) More details about image acquisition and processing should be given directly in the methodology section and not simply refer back to previous work.

2) P 6 of the supplementary material refers to "More 122 details on the image acquisition procedure are available in Supplemental Methods" - this is presumably an error.

3) Supplementary figure 1 seems to show a possible difference in the lead levels in men vs women. Is this difference significant and Is there any explanation for it?

Reviewer #3 (Remarks to the Author):

This is an interesting study carried out in 920 young adults to assess the effect of lead exposure on cognition, functional brain activity when performing a working memory task and microstructural properties using DTI measures. The paper is well written and the methodology used is suitable. The results allow to enrich the existing data for the identification of the effects of lead exposure on cognition and microstructural and functional properties of the brain
I have some comments and suggestions:

- Did the authors have information about lead exposure for each participant? Most participants seem represented in the low lead level. Did the authors find difference in site living, duration of lead exposure between participants (low / high hair lead level?)

- In the table with psychological / cognitive measures, not all participants were tested for all tests. It is unclear for example how 259 participants recruited for the study did not have the complex arithmetic task. What are the characteristics (demographic, lead hair level,..) of the participants excluded from these analyzes? Could the authors specify the missing data and is it the case in all analysis (such as in DTI and fMRI analysis)?

- By the absence of longitudinal data, it is indeed impossible to infer potential causal relationship.

However, did the authors attempt to conduct statistics to identify mediation pathways for example in the relationship between psychological / cognitive measures, microstructural volumes and hair lead level?

First of all, we would like to thank the editor and the reviewers for encouraging evaluations and constructive suggestions.

Other than the changes outlined below, we moved all the descriptions in supplemental methods to the methods section of the main text and made other minor modifications based on the format instructions of Communications Biology during revision. Moved parts are green colored and changed or added parts are red colored in the submitted manuscript.

We also corrected the y axis of Fig. 2 and Fig. 4 which included mistakes as follows.

Reviewers' comments:

Reviewer #1 (Remarks to the Author):

In their article, "Associations of lead exposure with functional and microstructural neural mechanisms", the authors explore the effect of lead on the brain using diffusion MRI and fMRI with a battery of various cognitive tasks results.

Thanks to a large dataset, this study shows the association between hair lead levels and changes in healthy young adults' structural and functional activity and cognitive measures. The article shows promising results to study the impact of lead exposure further. I have a few remarks and modifications that I think would help the reader better understand the methods and results. I also want to

note that I appreciate the prudence of the discussion regarding the results' causality. I think this is really important.

Major remarks:

Reviewer #1's major comment 1

Methods-Preprocessing:

Why did the authors use SPM8 to preprocess the data? This version is vastly outdated and requires using an obsolete version of Matlab as well. It exposes the analyses to potential bugs and much less accurate methods that nowadays standards. I think the authors should acknowledge this in the limitations.

We thank the reviewer for this comment.

We occasionally receive this question regarding the SPM version used for our DTI and fMRI analyses. We consulted the developer of robust rWLS and editor about this matter and we added the following point in the revised Supplemental Methods.

“Rationale for the use of SPM8 in the preprocessing of DTI/fMRI data and statistical analyses

Concerning preprocessing, we made use of SPM8 because our procedure is unique and has previously been validated with SPM8 (Takeuchi et al., 2013). Furthermore, when we use SPM12 and the same parameter sets that were validated in SPM8, apparent misclassifications of tissue types in certain brain areas repeatedly occur during the segmentation processes.

As for the 1st level analysis, we kept using SPM8's robust WLS's 1st level model specification & estimation, because result maps of SPM8's robust WLS's 1st level model specification & SPM8's robust WLS's estimation and those of SPM12's robust WLS's 1st level model specification & SPM12's robust WLS's estimation, are identical under default parameter settings. Further, for practical reasons, inconsistency of SPM version in the individual preprocessing and analytical procedures are not preferred in the review process.

As for the second level analysis, the distribution of t values from our results is identical regardless of SPM versions. As long as permutation procedures are taken, the results should not vary between SPM versions. Furthermore, our in-house script is only compatible with SPM8. “

The following figure has not been appended in the revised manuscript, but the following figure demonstrates the misclassification of our modified preprocessing procedures. The red line denotes occipital regions which incorrectly recognized dura matter as gray matter when SPM12 was used.

spm8

2 step segmentation
and complex DARTEL
procedures

resulting DARTEL template

spm12

2 step segmentation
and complex DARTEL
procedures

resulting DARTEL template

We did not mention this in the revised manuscript, but the results of SPM8's normal 1st model specification & estimation and those of SPM12 are identical under default parameter settings.

We also confirmed with the developer of robust WLS that there was nothing wrong in the fact that the results of SPM8's robust WLS' 1st level analysis and SPM12's robust WLS' 1st level analysis were identical, as core functionality did not change.

Reviewer #1's major comment 2

Statistical analyses of non-whole-brain analyses:

"Permutation analyses were conducted nine times for each cognitive measure, and the median p-value was used for analyses." Why nine times? Is there any explanation for that number?

We thank the reviewer for this comment.

Empirically speaking, conducting this R-based permutation only once sometimes leads to unstable results and choosing the median results among 9 times led to almost similar results no matter how many times we conduct the analyses. Since it does not take much time, we can perform larger numbers if it is deemed necessary by the reviewers or the editor.

Here, we simply added the following statement in the revised manuscript.

“(conducting 9 times lead to more stable results and the number was chosen empirically)”

Discussion:

Reviewer #1's major comment 3

"Lead accumulation results in reduced glutamine synthetase activity, which, in turn, causes glutamate accumulation and excitotoxicity, the effects of which are highly lethal to neurons. Lead is also known to cause oxidative stress via enhancing lipid peroxidation. These mechanisms may damage neurons or other tissue components to increase MD and reduce higher-order cognitive functions." This part gives a lot of hypotheses but could the authors provide some references to back them up? Also, even the rest of the paragraph only contains one reference, it would be helpful to have these mechanisms shown by other studies.

We thank the reviewer for this comment.

This part has been rewritten as follows to add supporting arguments and references.

*“This finding indicates the possible existence of mechanisms through which MD is increased whereas FA is unaffected. For example, lead accumulates in the mitochondria and eventually causes cell apoptosis (Lidsky and Schneider, 2003). Lead accumulation results in reduced glutamine synthetase activity, which, in turn, causes glutamate accumulation and excitotoxicity (**Lidsky and Schneider, 2003**), the effects of which are highly lethal to neurons. Lead is also known to cause oxidative stress via enhancing lipid peroxidation (**Halliwell and Gutteridge, 2015; Monteiro et al., 1985**). On the other hand, it is known that **cell-apoptosis in the brain leads to increased MD (Petrenko et al., 2018) and that oxidative stress is associated with increase of MD (or apparent diffusion coefficients)(Juang et al., 2014)**. Therefore, these mechanisms may damage neurons or other tissue components to increase MD and reduce higher-order cognitive functions.”*

Reviewer #1’s major comment 4

Figures:

Why figure 3 is overlaid on the single-patient image from SPM when the others are in the DARTEL space? DARTEL is the template used to register all the images, no?

We thank the reviewer for this comment.

All of the images are in the normalized space. The figure legend was rewritten in the following manner.

*“Regions with significant negative correlations between FA and hair lead levels are overlaid on the mean preprocessed (**including normalization**) but not smoothed, FA images of participants from whom the DARTEL template was created (**meaning this mean image is in the normalized space**).”*

The reason why we used this brain image in the presentation of the FA results is because, in the case of FA analyses, signal contamination from the gray matter areas is a matter of interest. By using this brain image, we can visually present how thoroughly the normalized image is aligned as well as how gray matter areas have been removed. But we can use the same template image that is used in the cases of other analyses, if that seems to be more appropriate to the editor and the reviewers.

Minor remarks:

Reviewer #1's minor comment 1

Statistical analyses of non-whole-brain analyses:

Please provide a more detailed description of the parameters. I reckon the ImPerm package offers multiple tools, and it would be easier for the readers to reproduce the analyses.

We thank the reviewer for this comment. Here, we simply used the following formula for each test, after loading the ImPerm library and the dataset. We added this formula in this subsection.

Result_x <- Imp(Test_x ~ sex + age + height + weight + BMI +

parents_education_level + family_income + hair_lead_level, datasetname_y,

seqs = TRUE)

summary(Result_x)

Reviewer #2 (Remarks to the Author):

Dear editor,

Thank you for inviting me to review this article. I read the work with interest. The article investigates the functional microstructural correlates of lead exposure.

While the principle aim behind this article is sound, I have certain concerns that I will elucidate in the following lists.

Major concerns:

Reviewer #2s major comment 1

1) It is commendable that the authors produce regression graphs in their figure. Unfortunately, the effects shown in figure one are very small. Therefore the authors should ensure that any claims that are made in the article are proportionate to the small effect sizes found in the analysis. Further, the authors

should explicitly give measurements of effect sizes in the text and explicitly comment on them.

We thank the reviewer for this comment.

First, we explicitly stated that the effect size was small wherever possible, unless the expression became unnatural across the revised abstract and discussion.

The following is the revised abstract.

*“Results revealed that greater hair lead levels are **weakly but significantly** associated with (a) increased working memory-related activity in the right premotor and pre-supplemental motor areas, (b) lower FA in white matter areas near the internal capsule, (c) lower MD in the dopaminergic system in the left hemisphere and other widespread contingent areas, and (d) greater MD in the white matter area adjacent to the right fusiform gyrus. Higher lead levels were also **weakly but significantly** associated with lower performance in tests of high-order cognitive functions, such as the psychometric intelligence test, greater impulsivity measures, and higher novelty seeking and extraversion. These findings reflect the **weak** effect of daily lead level on the excitability and microstructural properties of the brain, particularly in the dopaminergic system.”*

We reported effect sizes in beta for the psychological results and r for the imaging results. But, based on the sometimes observed opinion that part correlation coefficient is a more proper measure of effect size than r or beta, we ended up using part correlation coefficients consistently across the manuscript.

The explanation of the part correlation coefficients can be found below.

<https://effectsizefaq.com/2010/05/31/why-can%E2%80%99t-i-just-report-the-r-square-that%E2%80%99s-easy-enough-isn%E2%80%99t-it/>

Also, we added notes like following to explicitly explain the effect size for each of the results.

Part correlation coefficients for all of the significant associations were $< |0.14|$ and the effect sizes were weak.

We also added the following paragraph in the revised discussion.

“It has been shown that whole brain analyses tend to overestimate effect sizes, especially in small sample sizes³⁵ and that small sample studies tend to have greater effect sizes than meta-analyses³⁷. In our study, we recruited hundreds of subjects and observed relatively small effects sizes across measures ($|r| < 0.14$). However, due to the reasons described above, despite the relatively large sample size, the true effect size

between hair lead levels and psychological and imaging measure may still be smaller than what we reported. Previous meta-analyses also reported small effect sizes ($|r| < 0.16$) between lead levels and individual cognitive differences². However, these could be affected by measurement error when measuring lead and cognitive functions as well as low variances of lead levels among the normal population and not equal to low associations of brain mechanisms with cognitive mechanisms or lead levels. “

Reviewer #2s major comment 2

2) It is disappointing that the authors have chosen to use metrics such as FA, MD and other tensor scalars when much more advanced (and more biologically interpretable metrics are available - see NODDI, CHARMED, CSD fiber density etc ...). Further, it seems (unless I have misunderstood) that this work reuses a dataset from Takeuchi (2012). This is over 10 years ago and the quality of data available now is superior (32 directions at a b value of 1000 s/mm² is rather low). I would like to know why this dataset was chosen. This choice should be made explicitly in the manuscript. If on the other hand, this is not a reuse of data but a reuse of protocol, then I do not understand why the protocol was not updated. Perhaps I have misunderstood.

We thank the reviewer for this comment. The following points have been added to the limitation section of the revised Discussion.

“Lastly, this study is part of a long-standing project with data being collected gradually since 2008. Due to the nature of this study, structural scan protocols could not be updated and have remained fixed since the beginning. While this allowed us to have a good sample size, it limits the use of newer protocols such as neurite orientation dispersion and density imaging which could have added extra depth to the study. Future studies may need to investigate the effects of lead on brain function with more up-to-date imaging protocols.”

Reviewer #2s major comment 3

3) I do not understand why the software used for analysis (SPM) is not the most recent version, which has been available for many years now.

We thank the reviewer for this comment.

We occasionally receive this question regarding the SPM version used for our DTI and fMRI analyses. We consulted the developer of robust rWLS and editor about this matter and we added the following point in the revised Supplemental Methods.

“Rationale for the use of SPM8 in the preprocessing of DTI/fMRI data and statistical analyses

Concerning preprocessing, we made use of SPM8 because our procedure is unique and has previously been validated with SPM8 (Takeuchi et al., 2013). Furthermore, when we use SPM12 and the same parameter sets that were validated in SPM8, apparent misclassifications of tissue types in certain brain areas repeatedly occur during the segmentation processes.

As for the 1st level analysis, we kept using SPM8’s robust WLS’s 1st level model specification & estimation, because result maps of SPM8’s robust WLS’s 1st level model specification & SPM8’s robust WLS’s estimation and those of SPM12’s robust WLS’s 1st level model specification & SPM12’s robust WLS’s estimation, are identical under default parameter settings. Further, for practical reasons, inconsistency of SPM version in the individual preprocessing and analytical procedures are not preferred in the review process.

As for the second level analysis, the distribution of t values from our results is identical regardless of SPM versions. As long as permutation procedures are taken, the results should not vary between SPM versions. Furthermore, our in-house script is only compatible with SPM8. “

The following figure has not been appended in the revised manuscript, but the following figure demonstrates the misclassification of our modified preprocessing procedures. The red line denotes occipital regions which incorrectly recognized dura matter as gray matter when SPM12 was used.

spm8

2 step segmentation
and complex DARTEL
procedures

resulting DARTEL template

spm12

2 step segmentation
and complex DARTEL
procedures

resulting DARTEL template

We did not mention this in the revised manuscript, but the results of SPM8's normal 1st model specification & estimation and those of SPM12 are identical under default parameter settings.

We also confirmed with the developer of robust WLS that there was nothing wrong in the fact that the results of SPM8's robust WLS' 1st level analysis and SPM12's robust WLS' 1st level analysis were identical, as core functionality did not change.

Minor concerns:

Reviewer #2s minor comment 1

1) More details about image acquisition and processing should be given directly in the methodology section and not simply refer back to previous work.

Reviewer #2s minor comment 2

2) P 6 of the supplementary material refers to "More 122 details on the image acquisition procedure are available in Supplemental Methods" - this is presumably an error.

We thank the reviewer for these comments. Since Communications Biology prefers details about the methodology to be in the main manuscript rather than as a reference, we moved all the imaging procedures into the main text. We also removed the following sentence "More details on the image acquisition procedure are available in Supplemental Methods" in the revised manuscript.

Reviewer #2s minor comment 3

3) Supplementary figure 1 seems to show a possible difference in the lead levels in men vs women. Is this difference significant and Is there any explanation for it?

We thank the reviewer for this comment.

The figure may give that impression, but there are less females in this study and, "there were no significant sex differences in the logarithmic hair lead levels (two-tailed t-test, $p > 0.1$)." and this description has been added to the basic data subsection of the revised Results.

Reviewer #3 (Remarks to the Author):

This is an interesting study carried out in 920 young adults to assess the effect of lead exposure on cognition, functional brain activity when performing a working memory task and microstructural properties using DTI measures. The paper is well written and the methodology used is suitable. The results allow to enrich the existing data for the identification of the effects of lead exposure on cognition and microstructural and functional properties of the brain

I have some comments and suggestions:

Reviewer #3s major comment 1

- Did the authors have information about lead exposure for each participant? Most participants seem represented in the low lead level. Did the authors find difference in site living, duration of lead exposure between participants (low / high hair lead level?)

We thank the reviewer for this comment.

We believe the reviewer is asking if there is data of district-level lead (if hair lead level is not an indicator of lead exposure). Unfortunately, unlike the study of Marshall et al. (2020) (and like most of the studies of lead levels in the body using the general public sample instead of samples from the areas of particularly high lead exposure), we believe proper and up-to-date data is not available as far as we know. We can add such information if deemed necessary in the revised manuscript.

Reviewer #3s major comment 2

- In the table with psychological / cognitive measures, not all participants were tested for all tests. It is unclear for example how 259 participants recruited for the study did not have the complex arithmetic task. What are the characteristics (demographic, lead hair level,..) of the participants excluded from these analyzes? Could the authors specify the missing data and is it the case in all analysis (such as in DTI and fMRI analysis)?

We thank the reviewer for this comment.

First, *FA and MD analyses were performed with the data from 919 subjects after excluding data which had artifacts.*

Brain activity analyses were performed with data collected from 892 subjects after excluding data containing artifacts or improper behavioral data.

This text has been added to revised Results.

“The reason why the substantial portion of the subjects in the study have missing data in some of analyses of cognitive measures (Total intelligence score of TBIT, arithmetic tasks, reading comprehension) is because, in this long project, the measures that were gathered from subjects changed due to the limitation in the test time and many research purposes that are not related to hair analyses.

When there are missing data up to several subjects, then the reason is due to the misunderstanding of the rules despite administration of tests or failure to provide accurate answers to the questionnaires.”

This text has been added to Table notes of the revised Table 2.

Therefore, the biggest factor which influenced data omissions is the timing of data acquisition. Total intelligence score, arithmetic tasks, and reading comprehension differ in periods during which the data were not obtained. If needed, we can describe the timing of the period during which the measures were obtained or the relevant characteristics of the subjects during the period if that seems necessary, though the reason why those measures were not obtained during the period is not related to the project of hair analyses.

Reviewer #3s major comment 3

- By the absence of longitudinal data, it is indeed impossible to infer potential causal relationship. However, did the authors attempt to conduct statistics to identify mediation pathways for example in the relationship between psychological / cognitive measures, microstructural volumes and hair lead level?

We thank the reviewers for this comment.

Generally speaking, imaging measures in the significant areas of whole brain imaging analyses have strong overfitting effects towards the variables of contrast (Vul et al., 2009) and it is generally difficult to make secondary use of the results of whole brain analyses. We usually do not attempt secondary use of whole brain analyses unless specifically asked. Another theoretical difficulty is that neuroimaging measures are

indirect measures of certain neural function or integrity.

In our study, there are 7 significant psychological measures and 9 significant clusters.

We first conducted 63 partial correlation analyses between each pair of significant psychological measures and mean values of significant neuroimaging clusters. We not only corrected for covariates, we corrected for hair lead levels to remove the effects of overfitting in whole brain imaging analyses. We found no significant associations after correcting for multiple comparisons using the FDR method that was used in the manuscript (all $P > 0.2$, corrected). The results were attached as shown below.

However, it should be noted these analyses regressed the hair lead levels out and that these analyses are not sensitive when hair lead levels show true common covariance with psychological and imaging measures.

In addition, we sought to use predefined functional ROI for the suggested ROI analysis, as our previous study showed an association between premotor activity in the right premotor cortex and a measure of psychometric intelligence (Takeuchi et al., 2018).

Here, we sought to extract a functional ROI through an independent contrast ($T > 12$ in a one sample t-test comparing 2b-0b scores with activity) in the right premotor cortex.

However, after adjusting for covariates, beta contrast of this functional ROI did not significantly correlate with hair lead levels nor the total TBIT score, meaning it did not substantially overlap with the significant cluster (2-back – 0-back) of hair lead levels in this area. Therefore, we believe it is difficult to conduct the suggested analyses through the recommended method by leading experts in the field as we only have one data. We can add the following Table in the revised manuscript, if it is deemed to be necessary by the reviewer and the editor.

Table in the response to reviewer file. The partial correlation analyses between the mean values of significant clusters of hair lead level and significant psychological correlates of hair lead level. The upper line is the partial correlation coefficients and the lower line is uncorrected p values.

	FA (negative) cluster 1	FA (negative) cluster 2	MD positive cluster	MD negative cluster 1	MD negative cluster 2	MD negative cluster 3	MD negative cluster 4	2b - 0b positive cluster 1	2b - 0b positive cluster 2
Total intelligence score of TBIT	-0.034, 0.33	0.002, 0.949	-0.005, 0.887	0.030, 0.386	0.002, 0.951	0.02, 0.569	0.041, 0.238	-0.037, 0.296	-0.046, 0.192
Complex arithmetic	0.027, 0.489	-0.038, 0.333	0.012, 0.765	0.113, 0.004	0.067, 0.088	0.048, 0.222	0.058, 0.138	0.012, 0.768	0.071, 0.075
Reading comprehension	0.035, 0.310	-0.044, 0.204	-0.017, 0.627	0.031, 0.376	0.062, 0.076	0.051, 0.145	0.039, 0.267	0.061, 0.085	0.008, 0.813
Novelty seeking	-0.046, 0.164	-0.030, 0.373	-0.015, 0.641	-0.002, 0.958	-0.034, 0.301	-0.056, 0.090	0.002, 0.942	<0.001, 0.993	0.036, 0.283
Impulsiveness	-0.047, 0.160	-0.018, 0.588	-0.043, 0.197	-0.020, 0.537	-0.024, 0.477	-0.037, 0.263	-0.023, 0.493	0.034, 0.313	0.041, 0.225
Extraversion	-0.053, 0.110	0.006, 0.850	-0.014, 0.680	-0.006, 0.854	-0.045, 0.175	-0.089, 0.007	-0.036, 0.282	-0.021, 0.528	-0.014, 0.672
Cognitive reflectivity– impulsivity	0.042, 0.206	0.015, 0.662	0.008, 0.819	-0.017, 0.612	-0.002, 0.946	0.007, 0.843	0.016, 0.621	0.028, 0.413	-0.004, 0.905

Analyses of corrected all the covariates of corresponding whole brain imaging as well as hair lead level to exclude the associations caused by overfitting effects in whole brain analyses (Vul et al., 2009).

- Halliwell B, Gutteridge JM (2015) Free radicals in biology and medicine: Oxford University Press, USA.
- Juang C-L, Yang FS, Hsieh MS, Tseng H-Y, Chen S-C, Wen H-C (2014) Investigation of anti-oxidative stress in vitro and water apparent diffusion coefficient in MRI on rat after spinal cord injury in vivo with *Tithonia diversifolia* ethanolic extracts treatment. *BMC Complement Altern Med* 14:1-8.
- Marshall AT, Betts S, Kan EC, McConnell R, Lanphear BP, Sowell ER (2020) Association of lead-exposure risk and family income with childhood brain outcomes. *Nat Med* 26:91-97.
- Monteiro H, Abdalla D, Arcuri A, Bechara E (1985) Oxygen toxicity related to exposure to lead. *Clin Chem* 31:1673-1676.
- Petrenko V, Van De Looij Y, Mihailova J, Salmon P, Hüppi PS, Sizonenko SV, Kiss JZ (2018) Multimodal MRI imaging of apoptosis-triggered microstructural alterations in the postnatal cerebral cortex. *Cereb Cortex* 28:949-962.
- Takeuchi H, Taki Y, Thyreau B, Sassa Y, Hashizume H, Sekiguchi A, Nagase T, Nouchi R, Fukushima A, Kawashima R (2013) White matter structures associated with empathizing and systemizing in young adults. *Neuroimage* 77:222-236.
- Takeuchi H, Taki Y, Nouchi R, Yokoyama R, Kotozaki Y, Nakagawa S, Sekiguchi A, Iizuka K, Hanawa S, Araki T, Miyauchi CM, Sakaki K, Sassa Y, Nozawa T, Ikeda S, Yokota S, Daniele M, Kawashima R (2018) General intelligence is associated with working memory-related brain activity: new evidence from a large sample study. *Brain Struct Funct* Epub ahead of print.
- Vul E, Harris C, Winkielman P, Pashler H (2009) Reply to comments on "puzzlingly high correlations in fMRI studies of emotion, personality, and social cognition". *Perspect Psychol Sci* 4:319-324.

REVIEWERS' COMMENTS:

Reviewer #2 (Remarks to the Author):

I am satisfied that the authors have responded to all of my questions.

Reviewer #3 (Remarks to the Author):

All my comments have been addressed